# The Prediction of Clinical Mastitis in Dairy Cows Based on Milk Yield, Rumination Time, and Milk Electrical Conductivity Using Machine Learning Algorithms

**DOI:** 10.3390/ani14030427

**Published:** 2024-01-28

**Authors:** Hong Tian, Xiaojing Zhou, Hao Wang, Chuang Xu, Zixuan Zhao, Wei Xu, Zhaoju Deng

**Affiliations:** 1College of Science, Heilongjiang Bayi Agricultural University, No. 5 Xinyang Road, Daqing 163319, China; lijing8116@hotmail.com; 2Heilongjiang Provincial Key Laboratory of Prevention and Control of Bovine Diseases, College of Animal Science and Veterinary Medicine, Heilongjiang Bayi Agricultural University, No. 5 Xinyang Road, Daqing 163319, China; zzx032607@163.com; 3Animal Husbandry and Veterinary Branch, Heilongjiang Academy of Agricultural Science, Qiqihar 161005, China; tlwanghao777@126.com; 4College of Veterinary Medicine, China Agricultural University, No. 17 Tsinghua East Road, Haidian District, Beijing 100107, China; zhaoju2020@hotmail.com; 5Department of Biosystems, Division of Animal and Human Health Engineering, KU Leuven, Oude Markt 13, 3000 Leuven, Belgium; wei.xu@kuleuven.be

**Keywords:** cow, mastitis, prediction, rumination, electrical conductivity of milk, machine learning

## Abstract

**Simple Summary:**

Early monitoring and warning of mastitis in dairy cows in intensive farms in a timely manner are of great significance for protecting the welfare of cows, reducing the farms’ economic losses, and ensuring the quality and safety of dairy products. In this study, nine machine learning algorithms were used to predict naturally occurring clinical bovine mastitis pertaining to four specific stages of lactation. The Z-standardized dataset presents better results than the non-standardized ones. The multilayer artificial neural net (MNET) algorithm and random forest (RF) models are best suited for clinical mastitis prediction and management in farms. We also calculated the peak milk yield (PMY) of mastitic cows and that of healthy ones, and the former is higher than the latter. Overall, the results showed that machine learning algorithms can be applied to analyze real-time data obtained from intensive farms to develop an alerting system for the prediction of naturally occurring mastitis.

**Abstract:**

In commercial dairy farms, mastitis is associated with increased antimicrobial use and associated resistance, which may affect milk production. This study aimed to develop sensor-based prediction models for naturally occurring clinical bovine mastitis using nine machine learning algorithms with data from 447 mastitic and 2146 healthy cows obtained from five commercial farms in Northeast China. The variables were related to daily activity, rumination time, and daily milk yield of cows, as well as milk electrical conductivity. Both Z-standardized and non-standardized datasets pertaining to four specific stages of lactation were used to train and test prediction models. For all four subgroups, the Z-standardized dataset yielded better results than those of the non-standardized one, with the multilayer artificial neural net algorithm showing the best performance. Variables of importance had a similar rank in this algorithm, indicating the consistency of these variables as predictors for bovine mastitis in commercial farms with similar automatic systems. Moreover, the peak milk yield (PMY) of mastitic cows was significantly higher than that of healthy cows (*p* < 0.005), indicating that high-yielding cattle are more prone to mastitis. Our results show that machine learning algorithms are effective tools for predicting mastitis in dairy cows for immediate intervention and management in commercial farms.

## 1. Introduction

Clinical mastitis in early lactation can have negative impacts on the productivity of cows, including a temporary or permanent decrease in milk quality and production [1]. Mastitis significantly affects the welfare of animals at the individual or herd level and poses threats to the sustainability of dairy farming [2]. Prediction of mastitis at an early stage will help in successful disease intervention and reduce the risk of transmission of the pathogen, thereby potentially reducing the use of antibiotics [3,4,5]. Furthermore, early intervention could alleviate any pain or discomfort and hence increase the welfare of cows [6]. Measurement of somatic cell count (SCC) is the most frequently used method to indirectly evaluate subclinical and clinical mastitis at the herd or cow level [7,8]. In most farms, even in large commercial ones, the SCC in a cow or herd is normally tested once a month [9]. The electrical conductivity (EC) of milk, which is recorded for each milking, can also be used as a clinical predictor for mastitis of cows milked using rotary/robotic milkers [10,11,12,13]. There is a significant relationship between day rumination time and days relative to calving, twinning, subclinical hypocalcemia, subclinical ketosis, and retained fetal membrane [14]. Rumination and activity coupled with milk yield and body weight can be used to identify dairy cows with health disorders, such as mastitis, metritis, and lameness [15,16,17], ketosis [18], as well as the severity of inflammatory conditions evaluated using rumination time during the peripartum period [19].

Various commercially available sensors in intensive farms facilitate the collection and availability of a large amount of data, including estrus detection, synchronization protocol programming, and health management, which help in the efficient production and management of farms [20,21,22]. Recently, many researchers have explored the prediction or detection of subclinical/clinical mastitis in cows milked with automatic milking systems using classical machine learning algorithms such as support vector machine (SVM) [23,24], decision tree [25,26], random forest [25,27], gradient-boosted tree [25,27,28], Naïve Bayes [25,29], logistic regression [25,30], and neural networks [31,32,33,34,35,36]. Only a few studies on automatic milking systems have been conducted using random forest, Naïve Bayes, and extreme gradient [3] and SVM [37] algorithms to develop prediction or detection models on real-time data collected from milking parlors and/or management software adopted on the farms.

The deep learning algorithm has been used to detect key parts of the body of the dairy cow and the lameness of dairy cows in the Yangling district [38,39]. Data generated from precision dairy technology adopted by commercial farms have not been used previously for early warning, detection, or diagnosis of mastitis in dairy cows in Northeast China. This study aimed to verify the ability of machine learning algorithms to efficiently predict clinical mastitis in dairy cows based on real-time data of rumination time and physical activity generated from the monitoring collars, coupled with variables such as the EC of milk and milk yield from the rotary milking system.

## 2. Materials and Methods

This research was part of a large study aimed at developing a technology to improve farm management using big data, with a special focus on continuous monitoring, prediction, and precise detection of health disorders in lactating cows and calves in commercial herds in Northeast China using precision dairy technology and rotary/side by side milking system, along with environmental factors. Considering mastitis is more likely to occur in early productive life and lactation stages, cows were divided into subgroups according to their lactation stage, namely stage 1 (0–28 days in milk (DIM)), stage 2 (29–100 DIM), stage 3 (101–200 DIM), and stage 4 (201–305 DIM). We also accounted for parity (for mastitic cows, 25 cows with parity one, 63 cows with parity two, 117 cows with parity three, and 242 cows with parity ≥ 4, respectively; for healthy cows, 128 cows with parity one, 753 cows with parity two, 571 cows with parity three, and 694 cows with parity ≥ 4, respectively) to investigate the differences in forecast results. To obtain reliable results and improve decision-making by farm managers, data from healthy cows with similar DIM, daily milk yield, and the same parity as that of the sick ones were collected. All animal procedures were performed following the guidelines for the care and use of experimental animals at Heilongjiang Bayi Agricultural University (Daqing, China). The animal ethics committee of Heilongjiang Bayi Agricultural University approved the study protocol (FBD201603006).

### 2.1. Animal Housing and Feeding

We collected original data over 2.5 years (January 2020 to June 2022) from five commercial farms in Northeast China, the practical base of our university. The farms were located in three cities at latitudes and longitudes of 47.42 E to 51.03 E and 124.45 N to 129.18 N for the first city, 45.46 E to 46.55 E and 124.19 N to 125.12 N for the second city, and 44.04 E to 46.40 E and 125.42 N to 130.10 N for the third city.

Details about the environment of animal houses, collars worn by the cows, rotary milking systems, feeding patterns, and management modes used in the five farms are as reported previously [40]. The farms applied a total mixed ration to feed cows twice daily (500 h and 1300 h), with the feed pushed whenever necessary and fresh water made available at all times. They were milked thrice daily (300 h, 1100 h, and 1900 h) using a milking system (Data Flow, SCR Engineers Ltd., Netanya, Israel). The herds of these five farms were all monitored by neck collars (Collar, SCR Engineers Ltd., Netanya, Israel). Three farms adopted the Yimu Cloud management system (Yimu Technology Ltd., Beijing, China), and the other two used Data Flow Client management system (Data Flow, SCR Engineers Ltd., Netanya, Israel). Two farms raised 900~1000 cows per year on average during the period of experiment, with 1000~1200 cows, 1100~1200 cows, and 1600~1700 cows on the other three farms, respectively. Overall, the management modes and feeding patterns were similar among the considered herds.

### 2.2. Data Collection and Study Design

Information about the procedure of data collection and health-monitoring program is reported in detail in a companion manuscript [40]. The health-monitoring program was defined by our research team before the start of this study, and the farm staff (for each farm, 1 manager, 3 technicians, and 1 veterinarian with more than 15 years of experience monitoring cow health) were responsible for conducting the daily health monitoring of dairy cows. Clinical signs of mastitis were examined every three days by observing the udder and milk (i.e., hard quarter, heat or swelling, clots in milk, flakes, lumps, or clear/yellow milk) following calving until day 28 and were subsequently determined every seven days throughout lactation. Time from detection to diagnosis should not exceed 6 h, and details of animals, including cow identification number, quarter, date and time of diagnosis, and staff involved in the detection and diagnosis of disorders, were input to the management system software within 5 min after diagnosis.

In the present study, we monitored the weekly records of a total of 3031 healthy cows (without any disease during the experiment) and 587 cows suffering from naturally occurring clinical mastitis, with 685 mastitis events recorded from January 2020 to June 2022. The cows were initially grouped into two categories, mastitic cows and healthy cows, which were assigned a value of 1 and 0 when used as dependent variables, respectively, with the day of diagnosis and treatment considered as d-0, and the original variables were collected from seven (d-7) or three days (d-3) before diagnosis. After data preprocessing, the final data included information about parity, DIM, age at the time of disorders, milk yield, activity, six variables related to rumination time (daily rumination time, rumination at daytime, rumination at nighttime, the ratio of rumination time at daytime to that at nighttime, rumination deviation every 2 h, absolute values of the weighted rumination variation), and three variables related to the EC of milk (peak electrical conductivity of milk, daily percentage of the electrical conductivity of milk, standard deviation of the largest change in conductivity over the last three shifts) for a total of 2146 healthy cows and 447 mastitic cows (Table 1 and Appendix A).

### 2.3. Statistical Analyses

Statistical analyses were performed for all variables unless otherwise stated. The values of each variable in the interval (QL − 1.5 IQR, QU + 1.5 IQR) were used to conduct statistical analyses and to establish prediction models or were otherwise removed, where QL was represented as the lower quartile of each variable, QU was represented as the upper quartile, and IQR was represented as the upper quartile minus the lower quartile. After removing missing data and outliers, descriptive statistical tests were performed to characterize the measures of location and variability using means of frequency distribution tables and histograms; thereafter, the χ^2^ and *t*-tests were performed for categorical outcomes and continuous variables, respectively. Results were considered statistically significant at *p* < 0.05 (trends declared at 0.05 < *p* ≤ 0.10).

As the dataset used in this study to forecast mastitis was collected from five intensive farms that used the same collars, rotary milking system, and management system, the dataset was transformed using Z-standardization (that is, each variable was subtracted from the mean, and then divided by the standard deviation, such that the original values were mapped to an interval of [0, 1]) for stable and reliable generalization of the prediction model for the environment of other farms. Prediction models based on machine learning algorithms were applied to both the original and transformed datasets for the selection of the optimal prediction model.

### 2.4. Machine Learning Algorithms

We employed nine machine learning algorithms, including multilayer artificial neural net (MNET), binary logistic, SVM, Rpart, random forest, XGboost, AdaBoost, linear discriminant analysis (LDA), and Naïve Bayes using the R software version 4.1.2 (R Core Team, 2021, https://www.r-project.org/, accessed on 12 June 2022). For each adopted algorithm, “set seed ( )” was used to ensure the repeatability of our results, and we randomly divided data according to the dependent variable “Species” (binary variable “0” represented “healthy cows” vs. “1” “mastitic cows”) using the “createDataPartition” function. When performing each machine learning algorithm, a data subset consisting of 75% of the observations was selected as training data to construct the predicting models; the data subset consisting of the remaining 25% was used as testing data to assess the performance of the models. The parameters for the other eight machine learning algorithms used in this study were set as described previously [40].

The performance of each machine learning algorithm was assessed based on their sensitivity, specificity, accuracy, precision, Matthew’s correlation (evaluation indicator for the results of the binary classification model, especially for imbalanced category data), and area under the receiver operating characteristic (ROC) curve (AUC) value, which are defined as follows:$$Sensitivity=\frac{TP}{TP+FN},\;Specificity=\frac{TN}{TN+FP},\;Accuracy=\frac{TP+TN}{TP+TN+FP+FN},\;$$
$$Precision=\frac{TP}{TP+FP},MCC=\frac{TP*TN-FP*FN}{\sqrt{\left(TP+FP)(TP+FN)(TN+FP)(TN+FN\right)}}$$

For definitions of the abbreviations, see the companion article [40]. The description of the process of machine learning algorithms for developing the mastitis prediction model is shown in Figure 1.

## 3. Results

### 3.1. Variations in Variables of Milk Yield and Rumination Time

The approximate prevalence of clinical mastitis in the five commercial dairy farms ranged from 15 to 35% per 365 days during the experimental period. For the two farms that raised 900~1000 cows per year on average, the approximate prevalence of mastitis was 35% and 30%; for the farm with 1000~1200 cows, the approximate prevalence of mastitis was 25%; for the farm with 1100~1200 cows, the approximate prevalence of mastitis was 28%; and for the farm with 1600~1700 cow, the approximate prevalence of mastitis was 15%. Figure 2 depicts the general increasing trend of the six variables, which showed top importance in the trained prediction models. From d-3 to d-0, milk production was reduced in cows with mastitis, which is evident from the observed decreasing trend in the four stages (Figure 3A). For mastitic cows in the 0–28 DIM group, average daily milk showed higher variation (decreased from 38.82 ± 11.87 to 22.91 ± 12.32 kg/day) than in the other three stages. An increasing trend was observed for the ratio of daytime to nighttime rumination time (Figure 2A). For the cows in the 0–28 DIM group, the ratio of daytime to nighttime rumination time showed a larger variance (Figure 3B). This observation may be due to a sudden decrease in rumination during calving and an increase following calving in addition to mammary gland infection. Differences in rumination time every 2 h in mastitic cows in subgroups 0–28 and 201–305 DIM increased significantly from d-3 to d-0, whereas on d-2, cows in the subgroups 29–100 and 101–200 DIM showed a small decline (Figure 3C).

A noticeable increasing trend was observed in the absolute value of weighted rumination variation per 2 h from d-3 to d-0 (Figure 3D), with the largest variation on d-0, as expected. Significant differences (*p* < 0.001) in this variable were observed between the 0–28 and 29–100 DIM and 101–200 and 201–305 DIM subgroups.

### 3.2. Variations in Milk Electrical Conductivity

The largest variance for EC of milk (Figure 3E) was observed in mastitic cows in the 29–100 DIM subgroup, with a mean and standard deviation of 5.77 ± 0.58 mS/cm on d-3, 5.59 ± 1.13 mS/cm on d-2, 5.92 ± 0.58 mS/cm on d-1, and 6.04 ± 0.61 mS/cm on d-0. For all mastitic cows, the average values for EC of milk in the four subgroups were 5.77 ± 0.61 mS/cm on d-3, 5.72 ± 0.78 mS/cm on d-2, 5.88 ± 0.59 mS/cm on d-1, and 6.05 ± 0.65 mS/cm on d-0, respectively. The values on d-3 were significantly lower than that on d-0 (*p* = 0.021), d-2 (*p* = 0.014), and d-1 (*p* = 0.047). The daily percentage of EC change showed a gradually increasing trend (Figure 3F). There was a significant difference in the maximum change in conductivity over the last three shifts (Figure 3G) between mastitic cows at 0–28 DIM and the other three subgroups on d-3 and d-1 (*p* < 0.001).

### 3.3. Variations in Peak Milk Yield and Days in Milk

We calculated the peak milk yield (PMY) and DIM of PMY of mastitic and healthy cows. The PMY for all 447 mastitic cows was 5.60 ± 11.10 kg/day higher than that of the 2146 healthy cows (*p* < 0.005). The values of PMY of mastitic cows in the 0–28 and 29–100 DIM subgroups were 55.77 ± 4.23 and 55.79 ± 7.81 kg/day, respectively. The PMYs of mastitic cows in the 0–28 and 29–100 DIM subgroups were 9.06 ± 7.11 and 6.62 ± 10.44 kg/day higher than those of healthy cows (*p* < 0.005), while the difference in DIM of PMY between mastitic cows and healthy cows was not statistically significant.

### 3.4. Performances of Different Machine Learning Algorithms Using Non-Standardized Data

Due to the imbalance present in binary data (the number of healthy cows was four to five times higher than mastitic cows), we speculated that the accuracy may not be focused on the minority (mastitic cows). We used the Matthews correlation coefficient (MCC) to evaluate the metrics of the trained and tested models in the different machine learning algorithms.

Table 2 summarizes the performances of the nine machine learning algorithms based on non-standardized data for cows in the four specific lactation stages. For non-standardized data of cows at 0–28 DIM, the specificities of the nine machine learning algorithms varied from 85.59 to 94.59%, and the accuracy varied from 81.76 to 91.89%, while only the precision of the MNET algorithm exceeding 80% showed sensitivity. These results may be due to the high fluctuation of milk yield and large variation in the EC of milk and variables related to the rumination time of cows in this subgroup. As one of the model performance evaluation criteria, the MCCs of the nine machine learning algorithms were not high. The low precision signified that more healthy cows were provided to the veterinarians when the algorithms were developed into warning software. The overall performance of non-standardized data of cows in the 29–100 DIM subgroup was better than that of the 0–28 DIM subgroup, except for the performance of the LDA algorithm. Overall, the performance of MNET was better than that of the other algorithms, with all six criteria exceeding 0.80, with a specificity of 94.55% and an accuracy of 93.30%. Random forest had the second-best results for cows in this subgroup. For non-standardized data of cows in the 101–200 DIM subgroup, the sensitivity and precision of all nine algorithms were very low, although specificity was more than 80% (more than 90% for four algorithms). For cows in the 101–200 and 201–305 DIM subgroups, random forest showed better results than those of the other algorithms.

### 3.5. Performance of Different Machine Learning Algorithms Using Z-Standardized Data

Table 3 summarizes the performances of the nine machine learning algorithms using Z-standardized data for cows in the four specific lactation stages. The sensitivity of the XGboost algorithm increased from 1.52 to 13.26% when compared to that of non-standardized data. For data on cows in the 0–28 DIM group, XGboost and random forest performed better with similar results. Results of the Z-standardized data of cows in the four subgroups were better than those of non-standardized data. These results indicate that XGboost is better at modeling Z-standardized data of a large magnitude. For cows in the 0–28 DIM subgroup, the sensitivity, specificity, and accuracy of the MNET algorithm were higher than 90%, with the precision reaching 82.93%. Similarly, MNET showed a robust performance for cows in the 0–28 and 29–100 DIM subgroups, and the performance of random forest was the best for cows in the 101–200 and 201–305 DIM subgroups. Moreover, for cows in the subgroup 29–100 DIM, the specificities of six algorithms exceeded 90%, with the largest value of 96.73% for MNET, and the sensitivities of eight algorithms exceeded 80%, except for LDA. MNET performed the best among all the trained and tested models for cows in the subgroup 29–100 DIM, with an increase in sensitivity, specificity, and accuracy of 6.99, 7.14, and 7.07%, respectively, and the largest increase in precision of 17.22%. The Naïve Bayes algorithm showed a similar increasing trend in performance. The performance of the LDA algorithm showed improvement from that of non-standardized data; however, its performance was poorer than those of the other algorithms. Overall, the sensitivities of six models were higher for the 29–100 DIM subfamily than those in the other three stages, which suggests a possible application of the algorithm to develop a practical alert system for predicting mastitis in cows. Furthermore, nine out of nineteen variables of importance had a similar rank in the MNET algorithm in cows of the 0–28 and 29–100 DIM subgroups (Figure 4), indicating the consistency of these variables as predictors for mastitis of dairy cows in commercial farms with similar automatic systems. Parity did not have any significant effect on the occurrence of mastitis, which may be due to the small sample size or the variables involved in the algorithm.

## 4. Discussion

In commercial dairy farms, mastitis, a disease of the udder that is typically the result of bacterial infection, is associated with potentially increased use of antimicrobials and associated resistance, thereby affecting the welfare of dairy cows and increasing the rate of culling and/or death. Therefore, accurate and efficient mastitis prediction is valuable for timely intervention, leading to the protection of animal welfare both at individual and herd levels and ensuring associated food safety.

Relative to the traditional method, machine learning algorithms can model high-dimensional and noisy data efficiently, which have been successfully used to solve many biological problems, such as the prediction and detection of subclinical mastitis using various measurements gathered by automatic milking systems rather than laboratory tests, thereby reducing experimental costs, and without interfering with the daily working routine of the farmers or disturbing the cows.

The EC of milk is the measure of the resistance of a material to an electric current. Mastitis changes the blood capillary permeability [41]. For decades, this change in the conductivity of milk has been used as an indicator for clinical mastitis [42,43,44], with the frequency of use increasing in the dairy industry. Some studies have reported that EC exceeding 5.5 mS/cm could indicate subclinical mastitis [12,45]. In this study, the average value of the peak of EC for mastitic cows in all four subgroups exceeded 5.5 mS/cm, with an average value of 6.02 mS/cm on d-3, 5.99 mS/cm on d-2, and 6.01 mS/cm on d-1. Similar results were observed for milk yield, which was higher in mastitic cows than in healthy cows before the onset of mastitis [46]. In the early lactation period, high-yield dairy cows often experience relatively severe metabolic stress, and their self-immunity becomes relatively poor, which may give us a clue that cows at this stage are more likely involved in the risk of mastitis [47].

We also observed a delayed DIM in PMY in mastitic cows (62.51 ± 28.87 days) vs. healthy cows (55.17 ± 22.07 days), which was consistent with the results reported by Peiter et al. [48], where the DIM of PMY for mastitic cows in the 0–28 and 29–100 DIM subgroups were 61.5 ± 30.34 and 62.65 ± 27.02 days, respectively.

Consistent with the results reported by Stangaferro et al. [16] and King et al. [49], the daily rumination time, rumination time at nighttime, and daily milk yield started to decrease a few days before the diagnosis of mastitis. In line with our previous report [40], rumination deviation per 2 h, the sum of absolute values of the weighted rumination variation, and peak EC of milk showed a similar pattern of gradual increase before the diagnosis of mastitis. We also considered the daily percentage of change in the EC of milk and the standard deviation of the largest change in conductivity over the last three shifts (provided by the rotary milking system and uploaded to the management software), which showed significant differences between the sick and healthy cows, signifying that these two variables can be considered when developing a practical mastitis alert system. Although the variance from d-3 to d-0 in the change in EC of milk over the last three shifts did not show a noticeable trend as the absolute value of weighted rumination variation per 2 h, the two variables showed similar contributions to the forecast models.

In first-lactation Holstein-Friesian or Dutch Friesian cows, Barkema et al. [50] found that 30% of clinical mastitis cases occurred in the first 14 DIM. In first-lactation Iranian Holsteins, Moosavi et al. [51] found that although more clinical mastitis cases occurred in the first 74 DIM of lactation than later on, the duration of clinical mastitis was shorter when it occurred during this period. In this study, we divided the cows into four stages, as described in Section 2. For each machine learning algorithm, we presented the results derived using original non-standardized and Z-standardized data for four specific lactation stages to find the optimal models for predicting mastitis in cows one or three days before the actual onset. We also conducted experiments on logarithmic data and min–max normalization, which showed very poor results (sensitivity, specificity, accuracy; logistic regression, SVM, LDA, and Naïve Bayes algorithms were lower than 0.35). Using seven machine learning algorithms, Ebrahimi et al. [25] developed a prediction model of subclinical mastitis with several milking features measured using an automated monitoring system and SCC measured using an inline detector. They analyzed both non-transformed and Z-standardized datasets and found no significant differences between the two datasets. Ebrahimi et al. [25] designated EC as the most important parameter for predicting subclinical mastitis. The sensitivity of all their results exceeded 90%, whereas the specificity was lower than 50%. In our study, the specificities of all nine machine learning algorithms were higher than their sensitivities. Only the precision of the MNET algorithm modeled using Z-standardized data for cows in the 29–100 DIM group reached 90.91%, which indicated that, in future studies, we need to mine more reliable features to control the false positive rate and keep it as low as possible.

A previous study [52] has reported that the risk of clinical mastitis increased as parity increased, which may be because increasing parity increases the chances of infection with pathogens [53] in addition to the natural loss of teat defense mechanisms or immunity with an increase in years of service [54]. However, in the present study, parity was not confirmed as a significant indicator for naturally occurring mastitis. The activity of most cows at estrus was significantly affected; however, activity in mastitic cows in the four lactation stages did not display the expected difference nor give significant variance in any prediction model except in the MNET algorithm for data of cows in the 29–100 DIM subgroup.

Because of differences in the type of milking system used in the five farms, some parameters related to milk, such as its composition, temperature, or optical properties, were not included in the current experiment. The low accuracy and precision of the test model, the exclusion of data regarding milk parameters, and the smaller sample size of clinical mastitic cases pose challenges in the practical application of the prediction model. In future studies, in addition to the variables involved in the present study, other parameters should be mined, and algorithms with better performances should be explored to obtain more reliable and credible prediction outcomes.

Overall, these results indicated that models based on machine learning algorithms are useful tools in the development of prediction and/or detection models for mastitis in dairy cows monitored using collars and milked using a rotary milking system, which provides a broader understanding of some of the signs and symptoms of mastitis, leading to timely and efficient control and better management of this disorder.

## 5. Conclusions

We employed nine machine learning algorithms to analyze original data pertaining to 30 months and the corresponding Z-standardized dataset of four specific stages of lactation to train and test prediction models for the recognition and prediction of naturally occurring mastitis in dairy cows from five commercial farms. For cows in the 0–28 and 29–100 DIM subgroups, the MNET algorithm showed the best performance, and in the subgroups 101–200 and 201–305 DIM, random forest performed the best on both the original non-standardized and Z-standardized datasets. This comprehensive analysis establishes that machine learning algorithms can be applied to analyze real-time data obtained from intensive farms to develop an alerting system for the prediction of naturally occurring mastitis.

## Figures and Tables

**Figure 1 animals-14-00427-f001:**
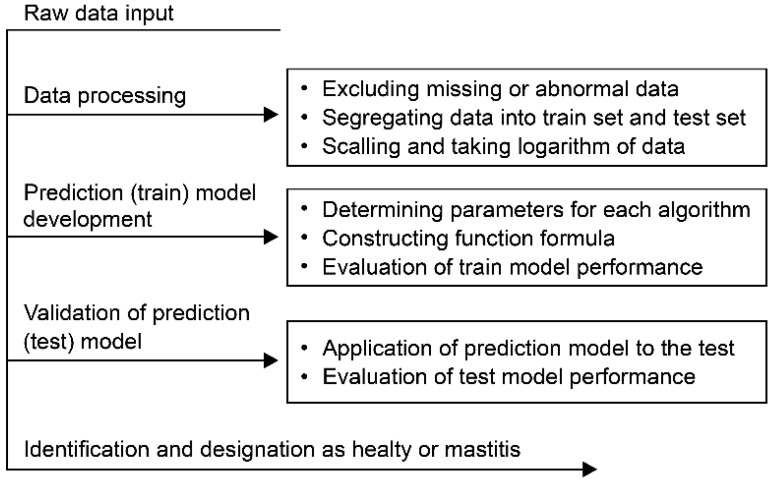
Data processing and construction of models.

**Figure 2 animals-14-00427-f002:**
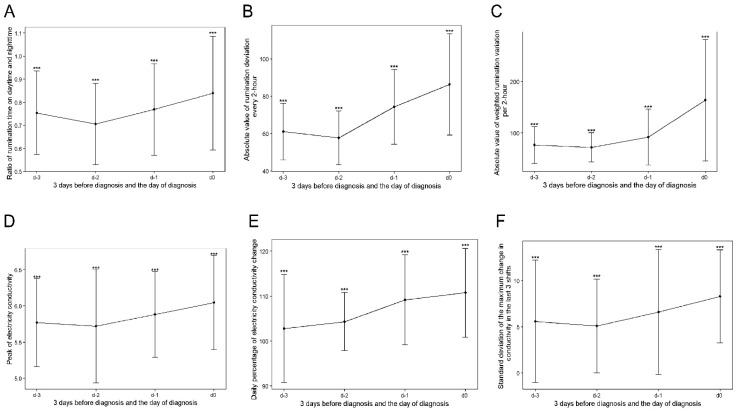
Line plots with error bars and significance of the top six variables with great importance in the trained models. Subgraphs (**A**–**F**) represent subgroups of variables of the ratio of rumination time during day and nighttime, absolute value of rumination deviation per 2 h, absolute value of weighted rumination variation per 2 h, peak of electrical conductivity of milk, daily percentage of variation in electrical conductivity of milk, and standard deviation of the largest change in conductivity over the last three shifts at the significance level of 0.001. The x-axis represents time from d-3 to d-0—that is, three days before diagnosis to one day before diagnosis—and the diagnosis day. “***” represents the difference between the two groups at a significance level of 0.001.

**Figure 3 animals-14-00427-f003:**
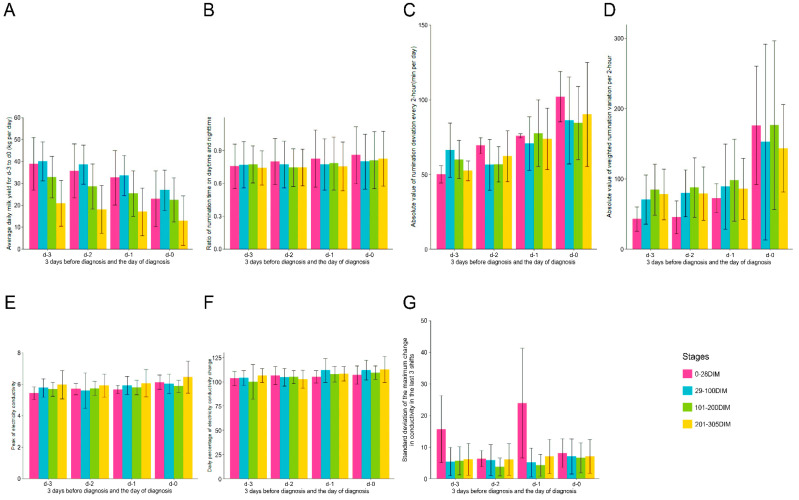
Bar plots with error bars of the variables involved in the training model on d-3 to d-0 for cows in the four subgroups. Subgraphs (**A**–**G**) represent subgroups of variables of average daily milk yield, ratio of rumination time at daytime to that at nighttime, rumination deviation every 2 h, absolute values of the weighted rumination variation, peak electrical conductivity of milk, daily percentage of change of the electrical conductivity of milk, and standard deviation of the largest change in conductivity over the last three shifts. The x-axis represents time from d-3 to d-0—that is, 3 days before diagnosis to 1 day before diagnosis—and the diagnosis day. The colors violet, blue, green, and yellow represent the subgroups “0–28 DIM”, “29–100 DIM”, “101–200 DIM”, and “201–305 DIM”, respectively. DIM, days in milk.

**Figure 4 animals-14-00427-f004:**
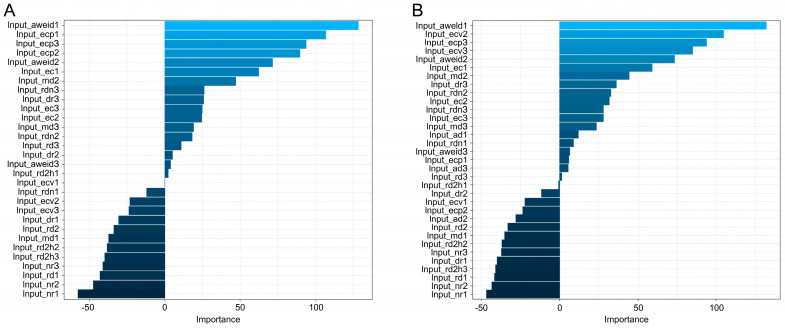
Variable importance obtained using a multilayer neural network (MNET) prediction model on data of cows at 0–28 DIM and 29–100 DIM is depicted in subgraphs (**A**,**B**).

**Table 1 animals-14-00427-t001:** Number of mastitic cows in the four subgroups.

	No.	No. of Cows	No. of Records of Physical Activity	No. of Records of Rumination Time	No. of Records of Electrical Conductivity of Milk
Subgroups					
0–28 DIM (14.7 ± 7.4)		74	592	3552	888
29–100 DIM (61.8 ± 21.1)		196	1568	9408	2352
101–200 DIM (141 ± 26.7)		111	888	5328	1332
201–305 DIM (254 ± 41.4)		66	528	3168	792
Total		447	3576	21,456	5364

DIM: days in milk.

**Table 2 animals-14-00427-t002:** Performances of the nine machine learning algorithms using non-standardized data for cows in the four specific lactation stages.

Metrics Subgroups Models	Sensitivity				Specificity				Accuracy				Precision				Matthews Correlation Coefficient				AUC			
	S1	S2	S3	S4	S1	S2	S3	S4	S1	S2	S3	S4	S1	S2	S3	S4	S1	S2	S3	S4	S1	S2	S3	S4
MNET	0.8378	0.8980	0.7857	0.7879	0.9459	0.9455	0.9302	0.8511	0.9189	0.9330	0.8947	0.8346	0.8378	0.8544	0.7857	0.6500	0.7838	0.8303	0.7159	0.6032	0.8965	0.8971	0.8721	0.7150
Logistic	0.7027	0.8469	0.6429	0.5455	0.8559	0.7964	0.8605	0.8404	0.8176	0.8097	0.8070	0.7638	0.6190	0.5971	0.6000	0.5455	0.5365	0.5856	0.4920	0.3859	0.7428	0.6628	0.6660	0.6055
SVM	0.6757	0.8163	0.6964	0.6061	0.8649	0.8909	0.8663	0.8511	0.8176	0.8713	0.8246	0.7874	0.6250	0.7273	0.6290	0.5882	0.5270	0.6826	0.5444	0.4528	0.7200	0.8073	0.6982	0.6529
Rpart	0.6486	0.7959	0.7321	0.7879	0.9279	0.9164	0.8953	0.8936	0.8581	0.8847	0.8553	0.8661	0.7500	0.7723	0.6949	0.7222	0.6065	0.7055	0.6167	0.6632	0.6951	0.8572	0.7714	0.7583
RF	0.6757	0.8980	0.8036	0.8182	0.9279	0.9091	0.9419	0.9043	0.8649	0.9062	0.9079	0.8819	0.7576	0.7788	0.8182	0.7500	0.6279	0.7730	0.7500	0.7030	0.8334	0.7644	0.8591	0.8325
XGboost	0.6757	0.6735	0.8030	0.8182	0.9189	0.9455	0.9244	0.9302	0.8581	0.8740	0.8908	0.8992	0.7353	0.8148	0.8030	0.8182	0.6121	0.6607	0.7274	0.7484	0.7741	0.7244	0.8914	0.8755
Adaboost	0.6757	0.7653	0.7321	0.7273	0.9279	0.8945	0.9128	0.8511	0.8649	0.8606	0.8684	0.8189	0.7576	0.7212	0.7321	0.6316	0.6279	0.6476	0.6449	0.5539	0.8031	0.7005	0.7755	0.7011
LDA	0.6757	0.5918	0.4643	0.6970	0.8829	0.8000	0.8314	0.7979	0.8311	0.7453	0.7412	0.7717	0.6579	0.5133	0.4727	0.5476	0.5537	0.3753	0.2975	0.4613	0.6160	0.5697	0.5153	0.6079
NB	0.7027	0.7857	0.8030	0.6364	0.8919	0.8400	0.8721	0.8511	0.8446	0.8257	0.8529	0.7953	0.6842	0.6364	0.7067	0.6000	0.5894	0.5883	0.6506	0.4784	0.7128	0.7064	0.7844	0.6660

S1, S2, S3, and S4 represent stage 1 (0–28 DIM), stage 2 (29–100 DIM), stage 3 (101–200 DIM), and stage 4 (201–305 DIM), respectively. AUC, area under the curve; DIM, days in milk; MNET, multilayer artificial neural net; SVM, support vector machines; RF, random forest; NB, Naïve Bayes.

**Table 3 animals-14-00427-t003:** Performances of the nine machine learning algorithms using Z-standardized data for cows at the four specific lactation stages.

Metrics Subgroups Models	Sensitivity				Specificity				Accuracy				Precision				Matthews Correlation Coefficient				AUC			
	S1	S2	S3	S4	S1	S2	S3	S4	S1	S2	S3	S4	S1	S2	S3	S4	S1	S2	S3	S4	S1	S2	S3	S4
MNET	0.9189	0.9184	0.8636	0.8485	0.9369	0.9673	0.9360	0.9063	0.9324	0.9544	0.9160	0.8915	0.8293	0.9091	0.8382	0.7568	0.8281	0.8828	0.7925	0.7281	0.9205	0.9182	0.8969	0.8400
Logistic	0.7027	0.8469	0.7273	0.7879	0.8739	0.8036	0.8663	0.8438	0.8311	0.8150	0.8277	0.8295	0.6500	0.6058	0.6761	0.6341	0.5622	0.5940	0.5808	0.5919	0.7150	0.6725	0.7099	0.7039
SVM	0.8649	0.8163	0.7424	0.8182	0.9099	0.8982	0.8721	0.8750	0.8986	0.8767	0.8361	0.8605	0.7619	0.7407	0.6901	0.6923	0.7441	0.6933	0.6013	0.6585	0.8533	0.7852	0.7108	0.7685
Rpart	0.8649	0.8061	0.7879	0.8182	0.9189	0.9200	0.9012	0.8854	0.9054	0.8901	0.8697	0.8682	0.7805	0.7822	0.7536	0.7105	0.7584	0.7192	0.6799	0.6734	0.8820	0.8682	0.7838	0.7887
RF	0.8919	0.8469	0.9242	0.8788	0.9279	0.9564	0.9535	0.9271	0.9189	0.9276	0.9454	0.9147	0.8049	0.8737	0.8841	0.8056	0.7932	0.8115	0.8660	0.7839	0.8371	0.8912	0.9035	0.8539
XGboost	0.8919	0.8061	0.8182	0.8485	0.9369	0.9236	0.9302	0.9167	0.9257	0.8928	0.8992	0.8992	0.8250	0.7900	0.8182	0.7778	0.8081	0.7251	0.7484	0.7443	0.8663	0.8769	0.8255	0.8633
Adaboost	0.7027	0.8265	0.7727	0.7273	0.9369	0.9345	0.9186	0.8854	0.8784	0.9062	0.8782	0.8450	0.7879	0.8182	0.7846	0.6857	0.6654	0.7586	0.6946	0.6012	0.7721	0.8509	0.8160	0.7200
LDA	0.6757	0.6735	0.6061	0.6970	0.9009	0.8109	0.8314	0.8542	0.8446	0.7748	0.7689	0.8140	0.6944	0.5593	0.5797	0.6216	0.5819	0.4584	0.4316	0.5317	0.6875	0.6208	0.6435	0.6527
NB	0.7027	0.8061	0.6970	0.7576	0.9189	0.9018	0.8895	0.9167	0.8649	0.8767	0.8361	0.8760	0.7429	0.7453	0.7077	0.7576	0.6335	0.6908	0.5893	0.6742	0.7726	0.7527	0.7148	0.7803

S1, S2, S3, and S4 represent stage 1 (0–28 DIM), stage 2 (29–100 DIM), stage 3 (101–200 DIM), and stage 4 (201–305 DIM), respectively. AUC, area under the curve; DIM, days in milk; MNET, multilayer artificial neural net; SVM, support vector machines; RF, random forest; NB, Naïve Bayes.

## Data Availability

The data presented in this study are available on request from the corresponding author. The data are not publicly available due to the privacy and confidentiality agreements as well as other restrictions.

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
