# Peer review of "The Prediction of Clinical Mastitis in Dairy Cows Based on Milk Yield, Rumination Time, and Milk Electrical Conductivity Using Machine Learning Algorithms"

_animals, 2024, doi:10.3390/ani14030427_

Round 1
Reviewer 1 Report
Comments and Suggestions for Authors
The manuscript is very needed in the industry. Overall, the manuscript is well written and easy to follow. I have a few minor comments for improvement.
Line 41-42: Why is this? Is it because the mastitis cows had longer DIM of PMY so they had higher peak milk?
Line 96-97: What are these numbers for parity? This is confusing and I don't understand.
Line 106: What is orignial data?
Line 112: How many cows per herd and what were the collars worn by cows? Briefly, add information.
Line 116: Is the EC from the SCR FreeFlow System?
Line 132-133-134-140: These numbers dont match with what is in Table 1. I don't understand why the numbers arent the same or I am confused. But shouldn't they match?
Line146: How did you remove data and outliers? What was the procedure? How did you determine outliers? How much data did you remove?
Line 186: How many cows per herd were these percentages based on?
Line 324-333: I don't think this paragraph is needed because it is materials and methods
Line 342-345: There must be an explanation for this here instead of just citing another study? Why does this happen?
Author Response
Dear Reviewer,
We greatly appreciate the helpful comments made by you. The response to your comments is given in the attachment of “Author to respond reviewer 1 point by point- MDPI”, uploaded to the system. We explained how we addressed each point given by you in the following.
Thank you very much for your comments.
We look forward to the good news.
We express our sincere thanks to you again.
Best regards
Chuang Xu
2024.01.23
Response to Reviewer 1 Comments
The manuscript is very needed in the industry. Overall, the manuscript is well written and easy to follow. I have a few minor comments for improvement.
Point 1: Line 41-42: Why is this? Is it because the mastitis cows had longer DIM of PMY so they had higher peak milk?
Response 1: Thanks for your detailed comments. The calculations were derived from the data of 447 mastitic and 2,146 healthy cows obtained from sensors in five commercial farms, actually, as we described that cows with high milk yield are more prone to mastitis, while, longer DIM of PMY didn’t contribute to the high yield. We have rewritten the DIMs and PMYs of mastitic cows and the healthy cows in the revised manuscript from Line 256 to 262, and Line 351 to 354.
Point 2: Line 96-97: What are these numbers for parity? This is confusing and I don't understand.
Response 2: Thank you very much for your kind remind. We have clarified the unclear statement “25, 63, 117, and 242 with parities of, 1, 2, 3, and ≥ 4” as “For mastitic cows, 25 cows with parity one, 63 cows with parity two, 117 cows with parity three, and 242 cows with parity ≥ 4, respectively. For healthy cows, 128 cows with parity one, 753 cows with parity two, 571 cows with parity three, and 694 cows with parity ≥ 4, respectively.” in Line 96 to 99 in the revised manuscript.
Point 3: Line 106: What is original data?
Response 3: The original data was from the sensors and management systems used by the farms, including daily activity, rumination time (daily rumination time, rumination at daytime, rumination at nighttime, the ratio of rumination time at daytime to that at nighttime, rumination deviation every 2 h, absolute values of the weighted rumination variation), daily milk yield, milk electrical conductivity (peak electrical conductivity of milk, daily percentage of the electrical conductivity of milk, standard deviation of the largest change in conductivity over the last three shifts), information of diagnosis of diseases (comprised the cow identification number, the date of diagnosis, the type of disorders, and the staff who detected and diagnosed the disorders, season, parity, age at the time of disorders, etc.). We have redescribed the original data in Line 138 to 142, and Line 146 to 152 in the revised manuscript.
Point 4: Line 112: How many cows per herd and what were the collars worn by cows? Briefly, add information.
Response 4: Thanks for your kind remind. We have added the information of neck collar, management system, milking system, cows per herd in the revised manuscript in Line 116 to 123.
Point 5: Line 116: Is the EC from the SCR FreeFlow System?
Response 5: Thank you very much for your detail reviewing. No, it isn’t. We made a mistake in the manuscript. EC is shorted for “electrical conductivity of milk“, which was monitored by the SCR Data Flow system used on the commercial farms, including three variables, namely, peak electrical conductivity of milk, daily percentage of the electrical conductivity of milk, standard deviation of the largest change in conductivity over the last three shifts. We have corrected the information in Line 117 in the revised manuscript.
Point 6: Line 132-133-134-140: These numbers dont match with what is in Table 1. I don't understand why the numbers arent the same or I am confused. But shouldn't they match?
Response 6: We greatly appreciate for your so detailed reviewing. Thank you very very much for pointing out the mistake we made on the values in the last column “No. of records of electrical conductivity of milk”, as this original variable was collected from three days (d-3) before diagnosis, the values presented in Table 1 were wrong. We have recalculated the values in the last column in Table 1 in the revised manuscript.
Point 7: Line146: How did you remove data and outliers? What was the procedure? How did you determine outliers? How much data did you remove?
Response 7: Thanks very much for your question. Yes, as you pointed out that we have added one sentence in Line 158 to 161 of the revised manuscript. Actually, at the beginning of the experiment, the weekly records of a total of 3,031 healthy cows (without any disease during the experiment) and 587 cows suffering from naturally occurring mastitis, with 685 mastitis events were monitored and downloaded from the systems used by the farms. The values of each variable in the interval (QL-1.5 IQR, QU + 1.5 IQR) were used to establish prediction models, otherwise deleted, where QL represented as the lower quartile of each variable, QU the upper quartile, and IQR the upper quartile minus the lower quartile. This process was conducted by functions in R software. Finally, records of a total of 2,146 healthy cows and 447 mastitic cows were analyzed.
Point 8: Line 186: How many cows per herd were these percentages based on?
Response 8: Thanks very much for your kind remind. We have added explanation for these percentages in Line 202 to 206 in the revised manuscript. During the period of experiment, for the two farms raised 900~1000 cows per year averagely, the approximate prevalence of mastitis was 35% and 30%, for the farm with 1000~1200 cows, the approximate prevalence of mastitis 25%, for the farm with 1100~1200 cows, the approximate prevalence of mastitis 28%, while for the farm with 1600~1700 cow, the approximate prevalence of mastitis 15%.
Point 9: Line 324-333: I don't think this paragraph is needed because it is materials and methods.
Response 9:Yes, this paragraph repeats the description in the materials and methods, hence, we have deleted those description in the revised manuscript.
Point 10: Line 342-345: There must be an explanation for this here instead of just citing another study? Why does this happen?
Response 10: As you pointed out, we added one sentence in Line 351 to 354 in the revised manuscript as “In the early lactation period, high-yield dairy cows often experienced relatively severe metabolic stress, and their self-immunity would become relatively poor, which may give us a clue that cows at this stage are more likely involved in the risk of mastitis” and one reference, and hence the order of some references have been changed correspondingly highlighted in red color.

Reviewer 2 Report
Comments and Suggestions for Authors
General Comments:
I have carefully reviewed the manuscript titled "Prediction of Mastitis in Dairy Cows Based on Milk Yield, Rumination Time, and Milk Electrical Conductivity Using Machine Learning Algorithms" submitted for publication in Animals. The manuscript presents research on mastitis detection in dairy cows, focusing on various methods, including sensor data, electrical conductivity, and machine learning techniques.
Introduction:
The introduction provides an adequate and comprehensive background on the topic of mastitis detection in dairy cows. It outlines the significance of the issue and its impact on milk production. It also presents previous studies about the use of machine learning algorithms for prediction and mastitis detection. The manuscript cites a substantial number of references related to mastitis detection, sensor technology, and machine learning applications.
Research Design:
The manuscript does not explicitly describe the research design, although it is reported that the manuscript is part of a large study about farm management using big data. The manuscript was developed according to a retrospective study, using data collected from five farms during 2.5 years.
Methods:
In general, the methods section is brief and provides sufficient details on how the research was conducted. However, it is essential to expand this section to include detailed information on:
a) Clarify the response variable evaluated (clinical or subclinical mastitis) and ensure consistent use of this terminology throughout the manuscript.
b) Specify how often clinical mastitis detection was performed during lactation. Was the frequency of clinical mastitis detection every 3 days along the lactation cycle? Clearly describe this process as it may affect the total number of clinical cases diagnosed.
c) Explain the criteria used to identify and remove outliers and provide details on the data editing process, including total raw data and data analyzed after outlier removal.
Results:
The manuscript adequately describes the main results of the study, including figures and tables. However, it is important to note that the difference reported in DIM (Days in Milk) of PMY (Peak Milk Yield) between mastitic cows and healthy cows (5.66 ± 29.55 days) is not statistically significant (p = 0.265). Thus, the results should not be described as different.
Additionally, the sentence "machine learning algorithms were not high" (Line 258-260) is vague and should be clarified for better understanding.
Conclusions:
In general, the conclusions are adequately described. However, the sentence (lines 411-412) "The calculated…" should be reviewed because this difference was not statistically significant.
Minor Comments:
Minor comments:
-
Specify whether the prediction is for clinical or subclinical mastitis in the title and throughout the manuscript (Line 1, Line 24, Line 57, Line 84, Line 134, Line 186, Line 399).
-
Clarify the sentence about DIM of PYM (Days in Milk of Peak Yield of Milk) for mastitic cows, as it seems the difference was not statistically significant (Line 42-44).
-
Correct to "at the herd or cow level" for clarity (Line 57).
-
Line 100-101: "All animal procedures….university". This sentence is duplicated.
-
Clarify the number of cows in the sentence: "...and 242 cows with…" (Line 98-101).
-
Adjust the sentence to "Milk production was reduced in cows with mastitis" (Line 189).
-
Line 215: "...error bars of the response variables…"
-
Line 359: "...Netherlands cows…" should be "Holstein cows"?
Author Response
Dear Reviewer,
We greatly appreciate the helpful comments made by you. The response to your comments is given in the attachment of “Author to respond reviewer 2 point by point- MDPI”, uploaded to the system. We explained how we addressed each point given by you in the following.
Thank you very much for your comments.
We look forward to the good news.
We express our sincere thanks to you again.
Best regards
Chuang Xu
2024.01.23
Response to Reviewer 2 Comments
General Comments:
I have carefully reviewed the manuscript titled "Prediction of Mastitis in Dairy Cows Based on Milk Yield, Rumination Time, and Milk Electrical Conductivity Using Machine Learning Algorithms" submitted for publication in Animals. The manuscript presents research on mastitis detection in dairy cows, focusing on various methods, including sensor data, electrical conductivity, and machine learning techniques.
Point 1: Introduction:
The introduction provides an adequate and comprehensive background on the topic of mastitis detection in dairy cows. It outLines the significance of the issue and its impact on milk production. It also presents previous studies about the use of machine learning algorithms for prediction and mastitis detection. The manuscript cites a substantial number of references related to mastitis detection, sensor technology, and machine learning applications.
Response 1: Thanks very much for your comments.
Point 2: Research Design:
The manuscript does not explicitly describe the research design, although it is reported that the manuscript is part of a large study about farm management using big data. The manuscript was developed according to a retrospective study, using data collected from five farms during 2.5 years.
Response 2: Thanks very much for your kind remind. We have added some information in Line 116 to 123, Line 131 to 134, Line 141 to 143 and Line 146 to 151 in the revised manuscript.
Point 3: Methods:
In general, the methods section is brief and provides sufficient details on how the research was conducted. However, it is essential to expand this section to include detailed information on:
- a) Clarify the response variable evaluated (clinical or subclinical mastitis) and ensure consistent use of this terminology throughout the manuscript.
- b) Specify how often clinical mastitis detection was performed during lactation. Was the frequency of clinical mastitis detection every 3 days along the lactation cycle? Clearly describe this process as it may affect the total number of clinical cases diagnosed.
- c) Explain the criteria used to identify and remove outliers and provide details on the data editing process, including total raw data and data analyzed after outlier removal.
Response 3:
- a) The dependent variable is categorical variable, and mastitc cow is assigned a value of 1, and the counterpart, the healthy one is assigned a value of 0. We have added one sentence in Line 141 to 143 in the revised manuscript.
- b) According to your kind remind, we have specified how often clinical mastitis detection was performed during lactation in Line 131 to 134 in the revised manuscript.
- c) According to your opinion, we have explained the criteria used to identify and remove outliers and provide details on the data editing process in Line 158 to 161 in the revised manuscript, and the total raw data and data analyzed after outlier removal see Line 152 and Table 1.
Point 4: Results:
The manuscript adequately describes the main results of the study, including figures and tables. However, it is important to note that the difference reported in DIM (Days in Milk) of PMY (Peak Milk Yield) between mastitic cows and healthy cows (5.66 ± 29.55 days) is not statistically significant (p = 0.265). Thus, the results should not be described as different.
Additionally, the sentence "machine learning algorithms were not high" (Line 258-260) is vague and should be clarified for better understanding.
Response 4: Thanks very much for your kind remind, we have rewritten the sentences in Line 256 to 262 in the revised manuscript. The sentence "machine learning algorithms were not high" have been rewritten in Line 275 to 278 in the revised manuscript.
Point 5: Conclusions:
In general, the conclusions are adequately described. However, the sentence (Lines 411-412) "The calculated…" should be reviewed because this difference was not statistically significant.
Response 5: Thanks for your detail comment, we have deleted the sentence.
Minor comments:
Point 6: Specify whether the prediction is for clinical or subclinical mastitis in the title and throughout the manuscript (Line 1, Line 24, Line 57, Line 84, Line 134, Line 186, Line 399).
Response 6: We have specified the prediction is for clinical mastitis in the title and throughout the manuscript (Line 2, Line 23, Line 26, Line 33, Line 56, Line 83, Line 140, Line 201, Line 408 in the revised manuscript).
Point 7: Clarify the sentence about DIM of PYM (Days in Milk of Peak Yield of Milk) for mastitic cows, as it seems the difference was not statistically significant (Line 42-44).
Response 7: Thanks for your kind remind, we have clarified this sentence in Line 39 to 43 in the revised manuscript.
Point 8: Correct to "at the herd or cow level" for clarity (Line 57).
Response 8: Thank you very very much for your so detail comment. We have corrected in Line 56 in the revised manuscript.
Point 9: Line 100-101: "All animal procedures….university". This sentence is duplicated.
Response 9: We greatly appreciate your reviewing. We have deleted the duplicated one.
Point 10: Clarify the number of cows in the sentence: "...and 242 cows with…" (Line 98-101).
Response 10: The value 242 is right. We have rewritten the sentence in Line 96 to 99 in the revised manuscript.
Point 11: Adjust the sentence to "Milk production was reduced in cows with mastitis" (Line 189).
Response 11: Thank you very much for your comment. We have adjusted the sentence in Line 209 in the revised manuscript.
Point 12: Line 215: "...error bars of the response variables…"
Response 12: These variables are all independent variables, not the response one, which is binary variable.
Point 13: Line 359: "...Netherlands cows…" should be "Holstein cows"?
Response 13: Thanks very much for your kind remind. We have corrected as Holstein-Friesian or Dutch Friesian in Line 372 in the revised manuscript.
